# Induction of Broad β-lactam Resistance in *Achromobacter ruhlandii* by Exposure to Ticarcillin Is Primarily Linked to Substitutions in Murein Peptide Ligase Mpl

**DOI:** 10.3390/microorganisms10020420

**Published:** 2022-02-11

**Authors:** Camilla Andersen, Migle Gabrielaite, Niels Nørskov-Lauritsen

**Affiliations:** 1Department of Clinical Microbiology, Aarhus University Hospital, DK-8200 Aarhus, Denmark; niels.norskov-lauritsen@rsyd.dk; 2Center for Genomic Medicine, Rigshospitalet, DK-2100 Copenhagen, Denmark; migle.gabrielaite@regionh.dk; 3Department of Clinical Microbiology, Odense University Hospital, DK-5000 Odense, Denmark

**Keywords:** cell wall recycling, derepression of β-lactamase, chronic infection

## Abstract

*Achromobacter* species are emerging pathogens in cystic fibrosis with inherent resistance to several classes of antimicrobial agents. We exposed strains with wild-type antimicrobial susceptibility to ticarcillin and generated mutants with broad β-lactam resistance. Within the detection limit of the assay, the capability to develop mutational resistance was strain-specific and reproducible. Mutational resistance was observed for all three tested strains of *Achromobacter ruhlandii*, for one of seven strains of *Achromobacter xylosoxidans*, and for none of five strains of *Achromobacter insuavis*. All mutants were resistant to piperacillin-tazobactam, while minimal inhibitory concentration of several other β-lactams increased 4–32-fold. Whole genome sequencing identified 1–4 non-synonymous mutations in known genes per mutant. All mutants encoded amino acid substitutions in cell wall recycling proteins, primarily Mpl, and the observed resistance is probably caused by hyperproduction of OXA-114-like β-lactamases. Related, but not identical substitutions were detected in clinical strains expressing acquired antimicrobial resistance.

## 1. Introduction

Cystic fibrosis (CF) is an autosomal recessive genetic disorder common in the Caucasian population [1]. The most serious threat to this group of patients is progressive lung destruction elicited by a vicious cycle of bacterial colonisation, infection, and inflammation [2]. *Achromobacter* species are emerging pathogens in CF [3,4]. Most patients with CF infected with *Achromobacter* spp. develop chronic infection, and longitudinal studies have reported cases of cross infection between patients [5,6,7]. The characterisation of isolates from the CF centre at Aarhus University Hospital in 2003–2014 showed the most common species to be *A. xylosoxidans* (36%), followed by *A. insuavis* (24%) and *A. ruhlandii* (19%) [8]; a similar trend was observed from the other Danish CF centre in Copenhagen [9]. Most isolates of *A. ruhlandii* in Denmark belong to the Danish Epidemic Stain (DES), a multi-drug resistant clone that has spread among patients at both centres.

*Achromobacter* spp. exhibits innate antibiotic resistance towards cephalosporins, aztreonam, fluoroquinolones and aminoglycosides, and antimicrobial treatment can be challenging due to additional acquired resistance in clinical isolates [4,10,11,12,13]. Intrinsic resistance mechanisms in *Achromobacter* spp. comprise multidrug efflux pumps and a chromosomal OXA-114-like β-lactamase [10,11,14,15].

The cell wall peptidoglycan polymer is the target of β-lactam antimicrobial agents. β-lactams disrupt the crosslinking of the peptidoglycan layer, which weakens the bacterial cell wall causing the bacteria to swell, and eventually burst and die. Inducible β-lactam resistance is well known from Gram-negative bacteria [16]. The regulatory protein AmpR can bind two ligands in a competitive manner, and AmpR normally restricts production of intrinsic AmpC β-lactamase to very low levels. Certain β-lactam antimicrobials liberate cell wall degradation products that act as AmpR-activator ligands. The regulatory protein undergoes conformational changes that disable its repression, and enhanced levels of AmpC are produced. When exposure to β-lactams ceases, AmpC production usually returns to baseline.

Gram-negative bacteria recycle a large proportion of the peptidoglycan components during growth and septation, and the mutational inactivation of proteins involved in recycling can cause the stable derepression of AmpC. AmpG is a permease that transports peptidoglycan degradation components to the cytoplasm, and AmpD is responsible for the cleavage of the degradation products (Figure 1); thus, AmpD reduces the presence of AmpR-activator ligands, while recycling can still be carried out. For *Enterobacterales*, the frequency of stable derepression of AmpC is caused by mutations in *ampD* > *ampR* > *ampG* [17]. The AmpR repressor ligand, UDP-muramyl-pentapeptide, is produced by the sequential addition of amino acids to UDP-muramyl substrate via five separate ligase enzymes, or by the murein peptide ligase Mpl, which ligates a pentapeptide directly onto the UDP-muramyl substrate. The pentapeptide is released by AmpD (Figure 1); consequently, the degradation of the AmpR activator ligand by AmpD is linked to generation of the AmpR repressor ligand by Mpl. Probably due to limited importance for clinical resistance in *Enterobacterales*, Mpl has received little attention. However, mutations in *mpl* can enhance β-lactamase production in laboratory-selected mutants and clinical isolates of *Stenotrophomonas maltophilia* and *Pseudomonas aeruginosa* [18,19,20,21].

In this study, we show that laboratory-induction of β-lactam resistance in *Achromobacter* spp. is linked to substitutions in enzymes responsible for AmpR ligands, particularly Mpl. Moreover, the induction is notably associated with the species *Achromobacter ruhlandii*.

## 2. Materials and Methods

### 2.1. Isolates

A total of 12 isolates of *Achromobacter* spp. were obtained from chronically infected patients with CF affiliated with the CF Centre at Aarhus University Hospital, Denmark during 2005–2014 [8], and one isolate of *A. ruhlandii* (non-DES related) was obtained from a patient affiliated with the CF Centre at Rigshospitalet, Copenhagen, Denmark (P4104_20141202, renamed CF02 in the present study [9]). Early isolates were tested if they expressed wild-type antimicrobial susceptibility, and only one stored isolate from individual patients were investigated. The type strains of *A. ruhlandii* (CCUG 38886^T^) (soil, 1998, https://www.ccug.se, (accessed on 16 January 2022)) and *A. xylosoxidans* (LMG 1863^T^) (ear discharge, 1971, https://bccm.belspo.be (accessed on 16 January 2022)) were also investigated. Genus-level identification was performed by Matrix-assisted laser desorption/ionisation time-of-flight analysis (MALDI Biotyper, Bruker, Bremen, Germany), and species-level by *nrd*A gene sequencing, as previously described [8,23]. Isolates were stored in semi-liquid agar (SSI Diagnostica, Hillerød, Denmark) at ambient temperature, and were cultured on 5% blood agar (SSI Diagnostica, Hillerød, Denmark) at 35 °C with 5% CO_2_.

### 2.2. Generation of Mutants

The selection of mutants and assessment of mutation frequency were performed in triplicates. Overnight cultures inoculated from single colonies were adjusted to an OD_600nm_ of ~1. 100 µL of suspension were spread on LB-agar containing 50 µg/mL ticarcillin (Sigma-Aldrich, Merck, Darmstadt, Germany), and 100 µL of 10^−6^, 10^−7^ and 10^−8^ dilutions were spread on blood agar for viable cell count. Plates were incubated for 72 h at 35 °C with 5% CO_2_ before assessment of growth. Mutants were categorised as true mutants if unrestricted growth was observed after subculture on ticarcillin-containing plates, and if they showed a specific resistance pattern with significant decrease in susceptibility towards piperacillin-tazobactam and meropenem, as evaluated by disk-diffusion tests. Mutation frequency was calculated as the number of ticarcillin-resistant colonies in proportion to the viable cell count, and the reported values correspond to the mean of three replicates.

### 2.3. DNA Sequencing and Analysis

Genomic DNA was extracted using the DNeasy Blood and Tissue Kit (QIAGEN, Hilden, Germany) following instructions for Gram-negative bacteria, but washing twice with buffer AW2. Whole genome sequencing included genomic library preparation using Illumina PCR Free Library Prep kit, followed by sequencing on the Illumina Novaseq 6000 platform (Illumina Inc., San Diego, CA, USA). Reads from the naive isolates were de novo assembled using SPAdes [24] and annotated into a GenBank file using either Prokka (https://vicbioinformatics.com/software.prokka.shtml (accessed on 14 October 2021)) [25] or Rapid Annotations using Subsystems Technology (RAST) (https://rast.nmpdr.org (accessed on 14 October 2021)). Genetic variation was determined by mapping the reads from mutant isolates to the naive genome using the Fixed Ploidy Variant Detection algorithm of CLC Genomics Workbench version 20.0.4 (QIAGEN, Hilden, Germany) with default settings. Single-nucleotide variants (SNV), deletions and insertions with a frequency of reads above 80% were considered for further analysis.

### 2.4. AmpD and Mpl Substitutions in Previously Published Genomes

A total of 92 genome sequences from clinical strains of *Achromobacter* spp. from the CF centre in Copenhagen were assembled as described [9]. With Snippy, version 4 [26] de novo assembled genomes were then mapped to the AmpD and Mpl gene sequences of *A. ruhlandii* (NCBI locus tags HV781_RS21220 and HV781_RS14840, respectively), and variant calling was performed.

### 2.5. Antimicrobial Susceptibility Testing

Standard antimicrobial susceptibility testing was performed by disk-diffusion technique on Mueller–Hinton agar (SSI Diagnostica, Hillerød, Denmark) according to the European Committee on Antimicrobial Susceptibility Testing (EUCAST) guidelines (www.eucast.org (accessed on 30 November 2021)).

Minimal inhibitory concentration (MIC) was determined for ten β-lactam antimicrobial agents using customised MIC plates for Gram-negative rods incubated for 18–24 h at 35 °C and analysed by the Sensititre^®^ Windows Software SWIN^®^ (Thermo Fisher Scientific, Waltham, MA, USA), according to manufacturer’s instructions.

## 3. Results

### 3.1. Isolates and Mutations

Seven strains of *A. xylosoxidans*, five strains of *A. insuavis* and three strains of *A. ruhlandii* were tested for induction of resistance to β-lactams by plating on 50 µg/mL ticarcillin. All three *A. ruhlandii* isolates (CF01, CF02 and CCUG), one of seven *A. xylosoxidans* isolates (CF03) and none of five *A. insuavis* isolates generated mutants with significantly reduced susceptibility to piperacillin-tazobactam and meropenem. An overview of the isolates included in the study is shown in Table 1. The induction of mutational resistance was thrice repeated with comparable results for mutated isolates. Mutation frequencies ranged from 0.4–3.8 × 10^−6^.

Genetic variation was determined by whole genome sequencing. The genomes of the four naive strains were assembled and annotated, and each of them were used as reference for reads from three separate mutational isolates, referred to as mut1-3. In addition, a clinical isolate with broad resistance cultured 10 years after the original, naive isolate, was also sequenced (CF01_2019). Table 1 shows the number of mutated genes (single nucleotide variations (SNVs) or indels) per isolate. For laboratory-generated isolates, between 1 and 15 mutated genes were disclosed, and the number of non-synonymous mutations in known genes ranged between 1–4/isolate. The clinical isolate CF01_2019 had accumulated a large number of mutations over a decade, with 38 non-synonymous variants occurring in known genes (Table 1). All laboratory-generated mutants as well as the clinical isolate encoded non-synonymous variants in either *mpl* or *ampD* (Table 2). Eleven isolates had mutations in *mpl*, with six being SNVs, plus three insertions and two deletions. One of the mutational isolates with broad resistance, *A. ruhulandii* CF01_mut1, encoded a non-synonymous SNV in *ampD* (p.Asp168Ala). The clinical isolate (CF01_2019) encoded a non-synonymous SNV in *ampD* (p.Glu124Lys). Five of eleven mutated Mpl genes encoded indels with frameshifts in the first half of the gene, thereby inactivating the ligase. The other six mutated Mpl genes were single amino acid substitutions occurring in widely separated regions of the protein (amino acid positions 111, 153, 236, 254, 305 and 340; Table 2).

### 3.2. β-lactam Susceptibility

The minimal inhibitory concentration (MIC) of mutated strains increased 8–16-fold for piperacillin-tazobactam, 4–8-fold for ertapenem, and 8–32-fold for meropenem (Table 3). Large increases were also observed for amoxicillin-clavulanic acid, aztreonam, cefotaxime, ceftazidime and ceftolozane-tazobactam, but these agents are rarely used for the treatment of *Achromobacter* spp. infections.

According to EUCAST Clinical Breakpoint Tables v.12.0 (2022) for *Achromobacter xylosoxidans*, all mutated strains were resistant to piperacillin-tazobactam (R > 4 mg/L), while only CF03_mut2 (plus the clinical isolate CF01_2019) was categorised as resistant to meropenem (R > 4 mg/L).

### 3.3. AmpD and Mpl Substitutions in Previously Published Genomes

We have previously reported genome sequences and results from routine antimicrobial susceptibility testing of 92 *Achromobacter* clinical isolates from the Copenhagen centre [9]. As a part of the present study, these sequences were mapped to the AmpD and Mpl gene sequences of *A. ruhlandii*. Two strains of *A. insuavis* expressed resistance to piperacillin-tazobactam and meropenem, but naive strains from the hosts were not investigated. From two patients (P3203 and P9403), strains of *A. xylosoxidans* with both wild-type and reduced susceptibility to β-lactams were sequenced, but differentiating mutations in AmpD or Mpl in the resistant isolates were not identified. With respect to *A. ruhlandii*, 27 DES isolates from 10 patients were sequenced, and a number of substitutions were noticeable in AmpD and Mpl. For AmpD, all DES isolates encoded p.Leu123Pro and p.Ala154Thr. These positions were not substituted in the unrelated non-DES *A. ruhlandii* strain P4104/CF02, nor were they detected in the laboratory-generated mutants, but p.Glu124Lys (CF01_2019) is a neighbouring substitution. For Mpl, all DES isolates encoded p.Thr241Met and p.Ala244Thr, which were not present in P4104/CF02. Neither of these substitutions were detected in the laboratory-generated mutants, but p.Trp236Arg and p.Asp254Gly (CF02_mut3 and mut2, respectively) are located in the proximity.

## 4. Discussion

This is the first study to report that β-lactam resistance in *Achromobacter* spp. is linked to substitutions in Mpl and AmpD genes by plating early isolates of different species with wild-type antimicrobial susceptibility on ticarcillin plates. *A. xylosoxidans*, *A. insuavis*, and *A. ruhlandii* are the common species of *Achromobacter* spp. cultured from patients with CF in Denmark, but the large majority of *A. ruhlandii* belong to the multi-resistant DES-clone that has spread among patients at both Danish CF centres. We could only identify two naive clinical strains of this species (plus the type strain), which were tested in parallel with seven strains of *A. xylosoxidans* and five strains of *A. insuavis*. Mutational testing was performed in triplicate on separate occasions with identical results, indicating that capability for mutational acquisition of broad β-lactam resistance was specific for individual strains (4 of 15 strains were repeatedly able). Moreover, there was a notable association with bacterial species, as all of three strains of *A. ruhlandii* were able, in contrast to one of seven *A. xylosoxidans* and none of five *A. insuavis*. This strain-specific capability to develop mutational resistance by exposure to ticarcillin is not clear; we have previously tested 90 *Achromobacter* spp. isolates from 42 patients with CF from our centre for mutational resistance to rifampicin, and absence of mutational competence for rifampicin resistance (detection limit 2 × 10^−9^) was rare [27].

Genome sequencing of 12 mutated strains only revealed a small number of non-synonymous mutations in known genes, and seven mutants only encoded substitutions in a single non-synonymous gene; however, cell wall recycling proteins AmpD or Mpl were mutated in all 12 isolates. By far, the most common target was *mpl*, which was mutated in 11 isolates. Five of the *mpl* mutations encoded indels with frameshifts, while six single amino acid substitutions were widely distributed in the protein. We did not quantitate expression of OXA-114-like β-lactamases in naive and mutated strains. The inactivation of Mpl diminishes the production of UDP-muramyl-pentapeptide, thereby allowing the competitor ligand to bind to AmpR. Thus, the enhancement of β-lactamase production is the evident explanation for the expression of phenotypic resistance in the mutant isolates. Although Mpl in *Enterobacterales* has received little attention, studies have shown that mutations in *mpl* can activate β-lactamase production in *S. maltophilia* and *P. aeruginosa* [18,19,20,21]. Calvopina and colleagues showed that clinical isolates of *S. maltophilia* with mutations in *mpl* expressed β-lactamase hyperproduction, as evaluated by nitrocefin hydrolysis [18]. Li and colleagues challenged *P. aeruginosa* with sub-lethal concentrations of ampicillin, *mpl* was frequently mutated in sequenced strains, and transcriptome analysis revealed a 7–9-fold upregulation of *ampC* [21]. In the present study, we also tested other β-lactam antimicrobials by plating on LB-agar containing 50 µg/mL piperacillin or ampicillin. However, these antimicrobials could not induce the level of β-lactam resistance as observed with ticarcillin (data not shown).

Apart from the resistant clinical isolate *A. ruhlandii* CF01_2019, isolated 10 years after the naive isolate, we were unable to identify decisive mutations in AmpD and Mpl in the few resistant strains of *A. xylosoxidans* that could be compared with their naive ancestors [9]. In that study, 29 isolates of *A. ruhlandii* were sequenced, 27 DES isolates cultured from 10 patients, plus 2 non-DES *A. ruhlandii*, including 1 strain with naive antimicrobial susceptibility (P4104/CF02). Despite the wide period of culture (2003–2018) and passage between 10 different patients, all isolates of DES encoded 2 conspicuous substitutions in AmpD (p.Leu123Pro and p.Ala154Thr), and 2 other in Mpl (p.Thr241Met and p.Ala244Thr). None of these substitutions were introduced in the laboratory-generated mutants, although some neighbouring substitutions were noted. Manipulation of the cell wall recycling proteins and quantitation of β-lactamase production in DES, the Danish epidemic clone of *A. ruhlandii*, must be addressed in future studies.

## 5. Conclusions

Exposure to ticarcillin induces mutational β-lactam resistance in susceptible strains of *A. ruhlandii*, rendering the mutants resistant to piperacillin-tazobactam, and with markedly reduced susceptibility to meropenem and other β-lactams. The capability to develop mutational resistance is rare in other *Achromobacter* spp. The antimicrobial resistance is associated with proteins of cell wall recycling and is probably caused by the overexpression of OXA-114-like β-lactamases.

## Figures and Tables

**Figure 1 microorganisms-10-00420-f001:**
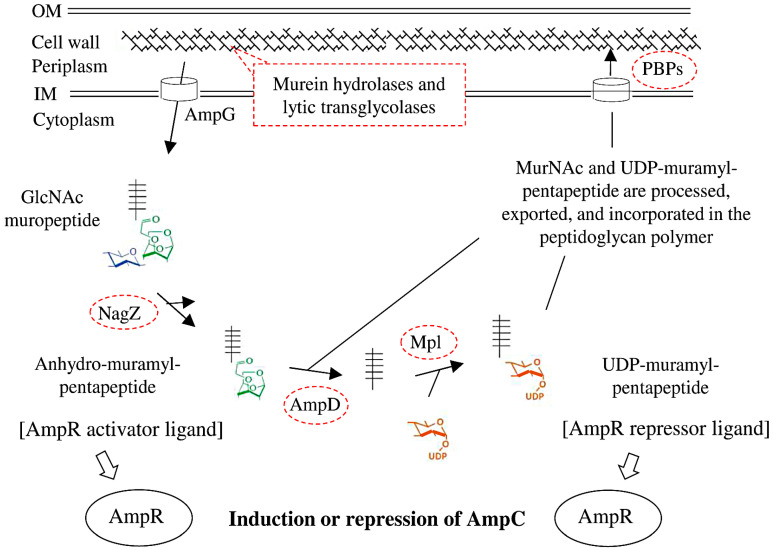
Schematic representation of cell wall recycling and β-lactamase expression in Gram-negative bacteria. The remodelling of the peptidoglycan polymer occurs during growth and septation, and lytic breakdown can be provoked by antimicrobials. Petidoglycan fragments (N-acetylglucosamine (blue)-anhydromuramyl (green)-pentapetide (black)) are liberated by lytic transglycosylase activity, transported to the cytoplasm through the AmpG permease, and hydrolysed by NagZ to create anhydromuramyl peptides. Under normal circumstances, the anhydromuropeptides are hydrolysed by AmpD, and through a series of reactions reused as cell-wall precursors. If anhydromuropeptides accumulate and bind to the regulatory protein AmpR, the production of AmpC β-lactamase is induced. The pentapeptide released by AmpD can be directly linked to UDP-muramic acid (orange) by mureine peptide ligase Mpl, or a pentapeptide can be assembled in successive steps (MurCDEF and Ddl). The Mpl product, UDP-Muramyl-pentapeptide, can be processed in the biosynthetic pathway and integrated in the nascent cell wall. UDP-Muramyl-pentapeptide also binds to the regulatory protein AmpR as a repressor ligand, decreasing the production of AmpC β-lactamase. (Based on references Johnson 2013 [16], Calvopiña 2018 [18], and Park 2008 [22]).

**Table 1 microorganisms-10-00420-t001:** Overview of isolates and number of mutated genes in mutational isolates.

Isolate	Species (Original Isolation) Induction	Non-Synonymous Mutations in Known Genes (Gene Symbols)	Non-Synonymous Mutations in Hypothetical Genes	Synonymous Mutations	Total
CF01_mut1	*A. ruhlandii* (2009, Aarhus) Ticarcillin ^1^	1 (*ampD*)	0	2	3
CF01_mut2		1 (*mpl*)	0	0	1
CF01_mut3		1 (*mpl*)	0	2	3
CF01_2019	*A. ruhlandii* (2019, Aarhus) Host factors ^2^	38 (*ampD, radA, gabR_4, cysS, thadh, Ion, kefF_1, fan1, est, argP_5, bigR, rpsI, yscL, algD_2, patA, sasA_10, rpsB, rstB_3, regA, oprB, sdgD_2, fusA_2, acsA_1, iclR_4, pcaJ_1, clsB_1, sasA_12, btrW, dnaA, amiD_2, lptF, dctD, livF_10, rstA, murG, ywrD_4*)	14	14	66
CF02_mut1	*A. ruhlandii* (2014, Copenhagen) Ticarcillin ^1^	1 (*mpl*)	0	1	2
CF02_mut2		2 (*mpl,* *ABC-trans*)	0	1	3
CF02_mut3		1 (*mpl*)	0	0	1
CCUG_mut1	*A. ruhlandii* (1998, soil, type strain; CCUG 38886^T^) Ticarcillin ^1^	4 (*mpl*, *dltA, mutL, clcD*)	4	2	10
CCUG_mut2		2 (*mpl*, *ihfB*)	2	2	6
CCUG_mut3		2 (*mpl*, *ihfB*)	2	1	5
CF03_mut1	*A. xylosoxidans* (2009, Aarhus) Ticarcillin ^1^	3 (*mpl*, *rspF, nosY*)	3	9	15
CF03_mut2		1 (*mpl*)	3	9	13
CF03_mut3		1 (*mpl*)	2	6	9

^1^ Ticarcillin: laboratory-generated mutant; ^2^ Host factors: late isolate from patient undergoing multiple courses of antimicrobial treatment.

**Table 2 microorganisms-10-00420-t002:** Characteristics of substitutions in *mpl* and *ampD* among mutational isolates.

Isolate	Gene	Type	Coding Region Change	Amino Acid Change
CF01_mut1	*ampD*	SNV	c.503A>C	p.Asp168Ala
CF01_mut2	*mpl*	SNV	c.1018A>C	p.Thr340Pro
CF01_mut3	*mpl*	SNV	c.331A>C	p.Thr111Pro
CF01_2019	*ampD*	SNV	c.370G>A	p.Glu124Lys
CF02_mut1	*mpl*	Insertion	c.45dupG	p.Leu16fs
CF02_mut2	*mpl*	Deletion	c.534delT	p.Arg180fs
CF02_mut3	*mpl*	Deletion	c.555delT	p.Asn185fs
CCUG_mut1	*mpl*	Insertion	c.538dupC	p.Arg180fs
CCUG_mut2	*mpl*	SNV	c.761A>G	p.Asp254Gly
CCUG_mut3	*mpl*	SNV	c.706T>C	p.Trp236Arg
CF03_mut1	*mpl*	SNV	c.458G>A	p.Arg153His
CF03_mut2	*mpl*	SNV	c.914T>A	p.Leu305Gln
CF03_mut3	*mpl*	Insertion	c.550dupC	p.Leu184fs

**Table 3 microorganisms-10-00420-t003:** Antimicrobial susceptibility testing of naive strains and mutational isolates (MIC).

Antimicrobial Agent		Amoxicillin-Clavulanic Acid	Aztreonam	Cefotaxime	Ceftazidime	Ceftazidime-Avibactam	Ceftolozane-Tazobactam	Ertapenem	Imipenem	Meropenem	Piperacillin-Tazobactam
Isolate	Gene										
CF01		≤4	4	2	≤0.5	≤0.5	4	≤0.12	2	≤0.12	≤1
CF01_mut1	*ampD*	64	>32	>8	4	≤0.5	>32	0.5	2	4	8
CF01_mut2	*mpl*	64	>32	>8	4	≤0.5	>32	0.5	2	4	8
CF01_mut3	*mpl*	32	>32	>8	4	≤0.5	32	0.5	2	2	8
CF01_2019	*ampD*	64	>32	>8	8	2	32	2	16	16	8
CF02		16	>32	>8	4	4	32	≤0.12	2	0.25	2
CF02_mut1	*mpl*	>64	>32	>8	8	4	>32	1	2	4	32
CF02_mut2	*mpl*	>64	>32	>8	8	4	>32	0.5	2	4	32
CF02_mut3	*mpl*	64	>32	>8	8	4	>32	0.5	2	4	32
CCUG		32	>32	>8	8	8	8	≤0.12	2	≤0.12	2
CCUG_mut1	*mpl*	64	>32	>8	8	8	>32	0.5	2	4	16
CCUG_mut2	*mpl*	>64	>32	>8	16	16	>32	1	4	4	32
CCUG_mut3	*mpl*	64	>32	>8	16	8	>32	0.5	2	2	16
CF03		8	>32	>8	16	16	>32	≤0.12	4	0.5	≤1
CF03_mut1	*mpl*	>64	>32	>8	16	8	>32	1	8	4	16
CF03_mut2	*mpl*	>64	>32	>8	16	16	>32	1	8	8	16
CF03_mut3	*mpl*	64	>32	>8	16	8	>32	0.5	8	2	16

## Data Availability

Mutated strains and genome sequences are available on reasonable request.

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
