# Peer review of "Induction of Broad β-lactam Resistance in Achromobacter ruhlandii by Exposure to Ticarcillin Is Primarily Linked to Substitutions in Murein Peptide Ligase Mpl"

_microorganisms, 2022, doi:10.3390/microorganisms10020420_

Round 1

Reviewer 1 Report

The paper entitled “Induction of broad β-lactam resistance in Achromobacter ruhlandii by exposure to ticarcillin is primarily linked to substitutions in murein peptide ligase Mpl” describes interesting findings about mutations identified in murein peptide ligase Mpl gene that confer resistance to β-lactamases. 

Major points:

1) The paper is not clear relative to the isolates investigated in the present study and comparison with other isolates from previously studies. This problem makes the paper difficult to read.

In particular, under material and methods section (lines 100-101), how many isolates were obtained from chronically infected patients with CF affiliated with the CF Centre at Aarhus University Hospital, Denmark during 2005–2014? And how many from Copenhagen? Please specify.

Line 103: “The type strains of A. ruhlandii (CCUG 38886T) and A. xylosoxidans (LMG 1863T) were also investigated.” Please insert the reference and the origin for the two strains.

Furthermore, the origin and the name of the isolates are not clear along the text and should be clarified and uniformed. For example, only at the end of the paper, under the discussion section it has been specified that P4104_20141202 has been “renamed CF02 when used for mutational testing in the present study”. This information should be explained under the material and methods section. Please, revise also lines 152-155, under the results section.

In addition, under table 1, the species of the isolates and the origin should be indicated.

The CF01_2019 strain is defined a clinical isolate along the text. Apart from the mutational isolates, are not the other naïve strains described in the paper clinical isolates? Please explain.

2) The paragraph “AmpD and Mpl substitutions in clinical strains” reports mainly the results from previously papers. I think it should contain first of all results of the present work and then comparison with other strain investigated in previously works.

3) Why do the authors suppose the observed resistance must be ascribed to hyperproduction of OXA-114-like β-lactamases, if they did not quantitate expression of OXA-114-like β-lactamases in naive and mutated strains? Cell wall recycling and Mlp correlate with production of AmpC β-lactamase in Gram-negative bacteria.

Minor points:

Specify also the breakpoint for ertapenem and how many strains are resistant.

Line 242, please correct singe with single.

Author Response

Dear Reviewer,

Thank you for taking the time to assess our manuscript.

We have addressed all the concerns that you have raised. Below, please find our response to your comments: Line numbers in [square brackets] refer to the revised version of the manuscript.

With kind regards,

Camilla Andersen

Major points:

1) The paper is not clear relative to the isolates investigated in the present study and comparison with other isolates from previously studies. This problem makes the paper difficult to read.

In particular, under material and methods section (lines 100-101), how many isolates were obtained from chronically infected patients with CF affiliated with the CF Centre at Aarhus University Hospital, Denmark during 2005–2014? And how many from Copenhagen? Please specify.

We have specified the number of isolates, and rephrased the section [L.104-110].

Line 103: “The type strains of A. ruhlandii (CCUG 38886T) and A. xylosoxidans (LMG 1863T) were also investigated.” Please insert the reference and the origin for the two strains.

We have inserted the reference, origin and website for the two type strains [L.110-112].

Furthermore, the origin and the name of the isolates are not clear along the text and should be clarified and uniformed. For example, only at the end of the paper, under the discussion section it has been specified that P4104_20141202 has been “renamed CF02 when used for mutational testing in the present study”. This information should be explained under the material and methods section. Please, revise also lines 152-155, under the results section.

We have tried to clarify the name of the isolates and mutants under the results section [L.163-168], in Table 1, and moved the information about CF02 to the material and methods section [L.108].

In addition, under table 1, the species of the isolates and the origin should be indicated.

We have expanded Table 1 with a new column showing strain characteristics. 

The CF01_2019 strain is defined a clinical isolate along the text. Apart from the mutational isolates, are not the other naïve strains described in the paper clinical isolates? Please explain.

The naïve strains are indeed clinical strains (in contrast to the type strains), and were chosen as early isolates from patients, who developed chronic infections. The laboratory-generated mutants, and the clinical strain isolated 10 years later (CF01_2019) express antimicrobial resistance, but are sub-clones of the same clinical strains (and the A. ruhlandii type strain). The focus of the study is to clarify the mechanism of resistance by identifying amino acid substitutions in resistant clones, compared with the naive strains. We have tried to clarify this difference in the revised manuscript in Table 1.

2) The paragraph “AmpD and Mpl substitutions in clinical strains” reports mainly the results from previously papers. I think it should contain first of all results of the present work and then comparison with other strain investigated in previously works.

The paragraph describes the search for similar substitutions in previously published genomes (and routine antimicrobial susceptibility testing) from the Copenhagen centre. This has been clarified in the material and methods section [L.144-145], and in the Results section [L.216-217]. Moreover, the section has been renamed [“AmpD and Mpl substitutions in previously published genomes”, L.213]

3) Why do the authors suppose the observed resistance must be ascribed to hyperproduction of OXA-114-like β-lactamases, if they did not quantitate expression of OXA-114-like β-lactamases in naive and mutated strains? Cell wall recycling and Mlp correlate with production of AmpC β-lactamase in Gram-negative bacteria.

We have corrected the statement to be more subtle [.. is probably caused by hyperproduction of OXA-114-like β-lactamases, L.30]. We plan to investigate the production of β-lactamase in our strains in future investigations.

Minor points:

Specify also the breakpoint for ertapenem and how many strains are resistant.

According to EUCAST, there are no clinical breakpoints for ertapenem, with respect to Achromobacter spp. or Pseudomonas spp.

Line 242, please correct singe with single.

We have corrected the typo.

Reviewer 2 Report

This is a study tested the competence for mutational resistance in A. ruhlandii ,  something that is known to be rare in other Achromobacter spp. The antimicrobial resistance is hereby associated through detection of SNPs and indels to be associated with proteins of cell wall recycling. The authors conclude that it is probably caused by overexpression of OXA-114-like 277 β-lactamases.

The importance of this study is that it is the first to report a link between beta-lactam resistance in Achromobacter spp. and substitutions in Mpl and AmpD genes

Major comments

The abstract is hard to understand and needs some editing.

For example, this sentence is confusing:

  • The first sentence is too complex and the connection to the rest of the paragraph is not clear.
  • "competence for mutational resistance"
  • Also in this sentence "… Twelve mutants 19 originating from four strains with wild type antimicrobial susceptibility were characterised." Are the four strains Achromobacter ruhlandii? Or are they another Achromobacter species? In the results it is clear that there are 3 ruhlandii and one

The introduction begins with a reference to CF – it would have been best if it was in the abstract – maybe something along the lines of "… Plating of the emerging CF pathogen Achromobacter…"

Line 128: There is no citation for SPADES, please add it.

Minor comments

Line 204 –".. both wild type- and reduced" : should this '–' be here?

Line 218-223:  While it is interesting to hear of the serendipitous nature of the change in your research, I do not believe this should be so extensively discussed, especially as no information regarding plasmids is shown.

Author Response

Dear Reviewer,

Thank you for taking the time to assess our manuscript.

We have addressed all the concerns that you have raised. Below, please find our response to your comments: Line numbers in [square brackets] refer to the revised version of the manuscript.

With kind regards,

Camilla Andersen

Major comments

The abstract is hard to understand and needs some editing.

For example, this sentence is confusing:

  • The first sentence is too complex and the connection to the rest of the paragraph is not clear.
  • "competence for mutational resistance"
  • Also in this sentence "… Twelve mutants 19 originating from four strains with wild type antimicrobial susceptibility were characterised." Are the four strains Achromobacter ruhlandii? Or are they another Achromobacter species? In the results it is clear that there are 3 ruhlandii and one

We have revised the abstract according to the concerns [L.16-24].

The introduction begins with a reference to CF – it would have been best if it was in the abstract – maybe something along the lines of "… Plating of the emerging CF pathogen Achromobacter…"

We have incorporated the term (emerging CF pathogen) into the abstract [L.16].

Line 128: There is no citation for SPADES, please add it.

The reference has been included (Bankevich Et al 2012. doi: 10.1089/cmb.2012.0021)

Minor comments

Line 204 –".. both wild typeand reduced" : should this '–' be here?

We have corrected the typo.

Line 218-223:  While it is interesting to hear of the serendipitous nature of the change in your research, I do not believe this should be so extensively discussed, especially as no information regarding plasmids is shown.

We have followed the reviewers advise and erased the paragraph from the paper.

Round 2

Reviewer 1 Report

The paper has been improved according to suggestions and the main points have been clarified.